# Assessment of S2S ensemble extreme precipitation forecast skill over Europe

Pauline Rivoire[1,2], Olivia Martius[1,3], Philippe Naveau[4], and Alexandre Tuel[1]

[1]Institute of Geography and Oeschger Centre for Climate Change Research, University of Bern, Switzerland
[2]Institute of Earth Surface Dynamics, Faculty of Geosciences and Environment, University of Lausanne, Switzerland
[3]Mobiliar Lab for Natural Risks, University of Bern, Switzerland
[4]Laboratoire des Sciences du Climat et de l'Environnement, ESTIMR, CNRS-CEA-UVSQ, Gif-sur-Yvette, France

**Correspondence:** Pauline Rivoire (pauline.rivoire@unil.ch)

**Abstract.** Heavy precipitation can lead to floods and landslides, resulting in widespread damage and significant casualties. Some of its impacts can be mitigated if reliable forecasts and warnings are available. Of particular interest is the sub-seasonal to seasonal (S2S) prediction timescale. The S2S prediction timescale has received increasing attention in the research community because of its importance for many sectors. However, very few forecast skill assessments of precipitation extremes in S2S forecast data have been conducted. The goal of this article is to assess the forecast skill of rare events, here extreme precipitation, in S2S forecasts, using a metric specifically designed for extremes. We verify extreme precipitation events over Europe in the S2S forecast model from the European Centre for Medium-Range Weather Forecasts. The verification is conducted against ERA5 reanalysis precipitation. Extreme precipitation is defined as daily precipitation accumulations exceeding the seasonal 95[th] percentile. In addition to the classical Brier score, we use a binary loss index to assess skill. The binary loss index is tailored to assess the skill of rare events. We analyse daily events locally and spatially aggregated, as well as 7-day extreme event counts. Results consistently show a higher skill in winter compared to summer. The regions showing the highest skill are Norway, Portugal and the south of the Alps. Skill increases when aggregating the extremes spatially or temporally. The verification methodology can be adapted and applied to other variables, e.g. temperature extremes or river discharge.

## 1 Introduction

Extreme precipitation is one of the most impactful weather-related hazards, in terms of loss of lives, economic impact and number of disasters (see e.g. the impact of storms and flood quantified in WMO, 2021). Additionally, if several extreme precipitation events occur in close succession (temporal clustering), flooding becomes more likely (Tuel et al., 2022). The successful mitigation of weather-related hazards depends on our ability to forecast them reliably. It is therefore crucial to quantify the skill of precipitation forecasts and improve the predictability of precipitation extremes for a better preparedness (Merz et al., 2020).

Subseasonal-to-seasonal (S2S) prediction refers to forecasting on timescales from about two weeks to a season. S2S prediction has a large range of applications (White et al., 2017, 2021), including the humanitarian sector, public health, energy, water management and agriculture. Forecast skill at this time scale is key to better manage natural hazards (Merz et al., 2020). S2S

predictions aim to fill the gap between weather forecasts and seasonal outlooks (White et al., 2017). Providing skillful pre-
dictions on subseasonal or monthly timescales is challenging (Hudson et al., 2011). Unlike short-range forecasts and seasonal outlooks that have been operational for many years, the S2S timescale was until recently a "predictability desert" (Vitart et al., 2012). The scientific community working with S2S forecasts has been growing rapidly (Mariotti et al., 2018; Merryfield et al., 2020; Domeisen et al., 2022). Many research organisations actively contribute to improving S2S forecast skill, for example the *Challenge to improve Sub-seasonal to Seasonal Predictions using Artificial Intelligence* (S2S-challenge, 2021).

Precipitation is a challenging variable to predict and, as a result, S2S forecasts of precipitation extremes have limited skill compared to other types of hazards (see e.g. case studies in  Domeisen et al., 2022; Tian et al., 2017; Endris et al., 2021). The analysis of S2S precipitation forecast skill could allow to identify regions and seasons with good or bad performance of the forecast. With this information, forecast users can know where and when the forecast information is useful or if it would require further improvement (with for example post processing, as in  Specq and Batté, 2020). Skill information is also useful to
identify potential sources of predictability and windows of opportunity (i.e. intermittent time periods with higher skill,  Mariotti et al., 2020). Most of the existing research on S2S prediction of precipitation extremes focuses on North America (Zhang et al., 2021; DeFlorio et al., 2019), Africa (de Andrade et al., 2021; Olaniyan et al., 2018) and Asia (Yan et al., 2021; Li et al., 2019). However little is known about the skill of S2S extreme precipitation prediction over Europe (Monhart et al., 2018; Domeisen et al., 2022). The present article aims to fill this gap.

S2S forecasts are ensemble forecasts that consist of several equally probable members, i.e. runs of the same numerical model with slightly different initial conditions (World-Climate-Service, 2021). Forecast skill is typically assessed with hindcasts. Hindcasts are forecasts run for past dates over sufficiently long time periods (about 20 years) to assess the quality of the forecast and to identify and correct model biases (e.g. Huijnen et al., 2012; Manrique-Suñen et al., 2020). The goal here is to quantify S2S forecast skill for extreme precipitation events over Europe using the forecast and hindcast data from the European
Centre for Medium-Range Weather Forecasts (ECMWF Vitart, 2020), one of the most frequently used and most skillful S2S modeling systems (de Andrade et al., 2019; Li et al., 2019; Stan et al., 2022; Domeisen et al., 2022).

Common metrics to evaluate the bias and the accuracy – and hence the skill – of ensemble forecasts include the mean absolute error, the probability integral transform, the interquartile range, the continuous ranked probability score (CRPS,  Hersbach, 2000; Gneiting et al., 2007; Crochemore et al., 2016; Monhart et al., 2018; Pic et al., 2022), the Brier score (Brier, 1950), and
the mean square skill score (Specq and Batté, 2020). However, these metrics capture the mean behavior of a variable: most are not directly suited to verify extreme events. The CRPS can be adapted to focus on extremes, using the threshold-weighted CRPS (Gneiting and Ranjan, 2011; Allen et al., 2021) or using extreme value theory (Taillardat et al., 2022). Another option to verify extreme events is the relative operating characteristic (ROC): it can be used to measure the ability of the ensemble forecast to identify above-normal precipitation events (Domeisen et al., 2022; Monhart et al., 2018). In this study, we transform
precipitation extremes into binary "threshold exceedance events", where the threshold is the daily precipitation $95^{th}$ percentile. The Brier score is usually employed to verify the binary forecast. However, it has limitations because of the unbalanced categories in our case. The extreme events dataset is composed of 95% zeros and 5% ones. A large part of the forecast and observation datasets are matching because of the large presence of "0s" (daily precipitation lower than the $95^{th}$ percentile)

in both datasets. To address this issue, we also use a binary loss index focusing on extremes ("1s"). We assess the extreme
events by proposing and using a simple extension of the binary loss score as introduced by Legrand et al. (2022) to ensemble
forecasts. This metric considers only the case of the occurrence of an extreme event in the forecast or in the observation or in
both but not the non-events (see section 2.3.2). This has the advantage that the score is not dominated by the correct prediction
of non-events. We compare our novel skill score to the classical Brier score (Brier, 1950). To overcome the double penalty
issue (i.e. when a location or timing error in the forecast is penalized by both a false-alarm and a missed event), we allow for
flexibility by aggregating the forecast information in spatial and temporal windows (Ebert et al., 2013).

This article is structured as follows. Section 2 contains a description of the forecast and verification data and the methods,
including the Brier score (Brier, 1950) and a binary loss index (adapted from Legrand et al., 2022). We present the results of
the analysis in Section 3. We discuss these results, draw conclusions and give an outlook in Section 4.

## 2   Data and Methods

### 2.1   Data

We use ECMWF's S2S precipitation hindcast data (cycle 47r2, , ECMWF, 2021; Vitart, 2020; ECMWF, 2022a) from 2001 to
2020. It is composed of 11 ensemble members, initialized twice a week and run for 46 days. We focus on Europe, in the spatial
box [30°N ; 72°N] × [-15 °E; 49.5 °E]. The hindcast period covers 20 years with 2080 forecast initializations between 2001-
01-04 and 2020-12-30 (twice a week, on Monday and Thursday). The data were downloaded at the model spectral resolution
O320 (ECMWF, 2022b, c) and regridded for the analysis to a 0.5°×0.5° regular grid using a first-order conservative remapping
(Jones, 1999; CDO, 2018).

ERA5 precipitation (Hersbach et al., 2019) is used here as the verification dataset. The choice of a reanalysis dataset is
motivated by its continuous spatial and temporal availability and to avoid the uncertainties due to the inherent spatial sparsity
of weather station networks (Hofstra et al., 2009; Rivoire et al., 2021). Daily precipitation are extracted over the same time
period, from 2001-01-04 to 2020-12-30 plus 46 lead time days i.e. 2021-02-14, with a spatial resolution of 0.5°×0.5°. For the
sake of simplicity, "observation" refers to ERA5 in the remainder.

### 2.2   Definition of extreme events

We define precipitation extremes as binary exceedances of daily precipitation accumulation above its $95^{\text{th}}$ seasonal all-day
percentile (i.e. over all days in March-April-May –MAM–, June-July-August –JJA–, September-October-November –SON–
or December-January-February –DJF–). Figure A1 in the appendix shows the $95^{\text{th}}$ percentile ($Q_{95}$) in ERA5 in Europe, for
the period from 2001-01-04 to 2021-02-14. For the hindcast data, we also compute percentiles separately for each lead time:
for a given lead time day and a given season, $Q_{95}$ is computed from daily precipitation of all the ensemble members pooled
together. Figure 1 shows the bias in this percentile between the forecast and ERA5 data for four different lead times. In this
figure and all the following ones, only values at grid points where $Q_{95}$ in the observations is greater than 5mm per day are

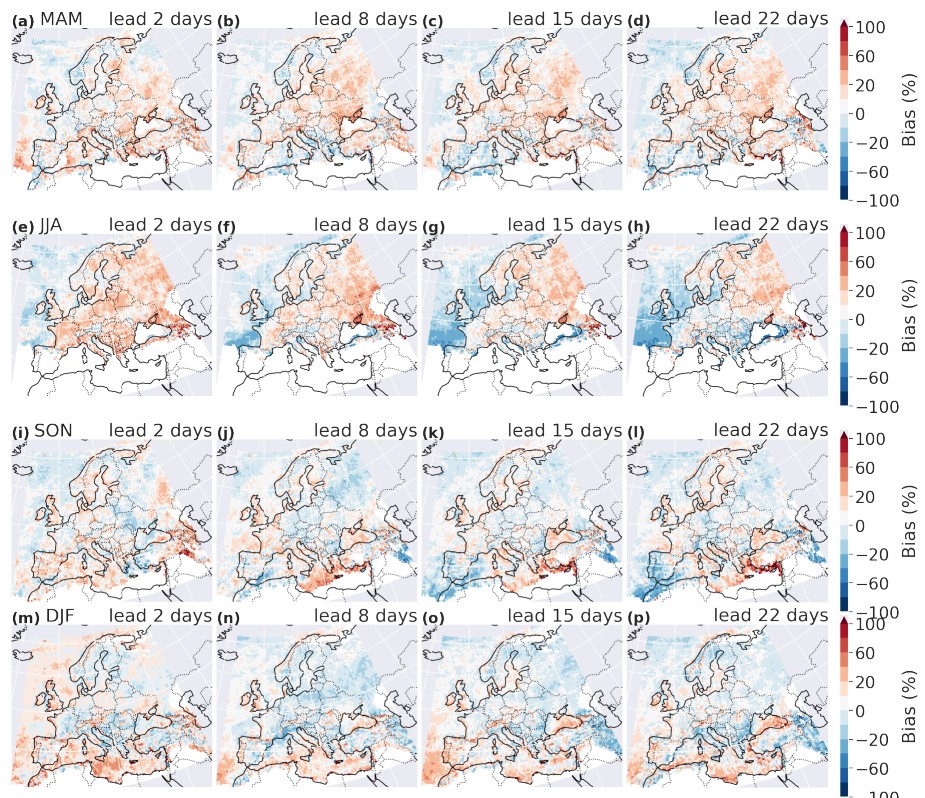

**Figure 1.** Forecast bias in the $95^{\text{th}}$ percentile ($Q_{95}$) compared to ERA5, for spring (MAM, (a)-(d)), summer (JJA, (e)-(h)), autumn (SON, (i)-(l)) and winter (DJF, (m)-(p)) at a 2-day (first column), 8-day (second column), 15-day (third column) and 22-day (last column) lead time. Grid points with $Q_{95} \leq 5$mm per day are displayed in white.

shown. For a lead time of one day, the forecast generally underestimates $Q_{95}$. For lead times between 2 and 46 days, some regions have a positive bias (central Europe in spring and summer) and some have a negative bias (the Alps in summer, autumn and winter, Norway in spring, autumn and winter, see Figure 1). Generally over Europe, the bias depends on the lead time and on the season. However, the bias over oceans often has the opposite sign of the bias over land.

## 2.3  Metrics

We use the Brier score and a binary loss index to assess the forecast skill in extreme events. We compute the Brier score and the binary loss index for the extended winter season (NDJFMA, i.e. November to April) and extended summer season (MJJASO, i.e. March to October). When defining the extremes (see previous section) we used 3-month long seasons because of the strong seasonal cycle in extreme precipitation (see figure A1). The choice of extended seasons for the skill analysis is a compromise between having enough extreme events for a robust analysis and capturing the seasonality of the forecast. As consequence

the probability of the extreme events is no longer exactly 0.05 if extreme events are not homogeneously distributed within the MAM and SON seasons.

### 2.3.1   Brier Score

The Brier score $B$ is defined as the mean square difference between forecast probability and binary observations (Brier, 1950):

$$B = \frac{1}{n_D} \sum_{i=1}^{n_D} (f_i - Y_i)^2,$$

where $n_D$ is the total number of days (i.e. the number of initializations in the given extended season: about 1040 per lead time, i.e. half the number of initializations per year); $Y_i$ the binary observation of extreme for day $i$ ($Y_i=1$ if the daily precipitation

exceeds the $95^{\text{th}}$ and $Y_i = 0$ otherwise); $f_i$ is the forecast probability of extreme occurrence for day $i$, i.e. the mean of the ensemble members: $f_i = \frac{1}{M} \sum_{m=1}^{M} F_{(i,m)}$ , with $M$ the number of ensemble members (here $M = 11$) and $F_{(i,m)}$ the binary forecast for a given ensemble member $m$ for day $i$.

     $B$ is negatively oriented (the lower the better). The climatological Brier score $B_{\text{clim}}$ is used as a reference value for the skill calculation:

$$B_{\text{clim}} = \frac{1}{n_D} \sum_{i=1}^{n_D} (p - Y_i)^2,$$

where $p$ is the climatological extreme event probability. Note that the value of this probability is not exactly 0.05, as two of the 3-month seasons are split to form the extended seasons. $p$ is therefore computed empirically.

     The forecast is skillful if its Brier score is lower than the climatological Brier score. These scores can be compared using the Brier Skill Score ($BSS$):

$$BSS = 1 - \frac{B_{\text{hind}}}{B_{\text{clim}}}.$$

$BSS$ varies between $]-\infty; 1]$ and is positively oriented (the closer to one, the better). For a given lead time day, a forecast has skill if $BSS > 0$. From here on, the expression "the last skillful day" refers to the largest lead time day with skill.

### 2.3.2   Binary loss index

Legrand et al. (2022) studied in detail a risk function defined as the ratio between the empirical probability of having an extreme event in either the observation dataset or the forecast dataset, and the empirical probability of having an extreme event in the observations or the forecast (including having an event in both datasets). In our context, the risk function can be written:

$$R^{(u)}(X) = \frac{\mathbb{P}(X^{(u)} \neq Y^{(u)})}{\mathbb{P}(Y^{(u)} = 1 \text{ or } X^{(u)} = 1)}$$

     where $Y^{(u)}$ is the binary observation, $Y^{(u)} = 0$ (resp. $Y^{(u)} = 1$) if the observed daily precipitation is lower (greater) than a certain threshold $u$; and $X^{(u)}$ is the binary forecast $X^{(u)} = 0$ (resp. $X^{(u)} = 1$) if the predicted daily precipitation is lower

(greater) than $u$.

     The risk function $R^{(u)}$ focuses on how well the "1" values (extreme event days) match between observation and forecast. It does not take into account steps when neither the forecast nor the observation experience an extreme event. $R^{(u)}(X)$ varies

between $[0; 1]$ and is negatively oriented (the closer to zero, the better the forecast is). The strength of $R^{(u)}(X)$ is its asymptotic behavior: even for very rare events, both the over-optimistic and over-pessimistic forecasts will be penalized. In other words, even for very large threshold $u$, i.e. $Y = 1$ for very rare occasions (but at least once), if the forecast is too optimistic and $X = 0$ for all time steps, then $R^{(u)}(X) = 1$ ("naive" classifier, Legrand et al., 2022). A very pessimistic forecast will be penalized the same way ("crying-wolf" classifier, see Legrand et al., 2022). The commonly-used Brier score rather assesses the average behavior, with a very weak penalty for under-represented classes. Because all days are compared, the assessment of rare extreme events (missed, false alarm or hit) by the Brier score is lost among the huge amount of correctly predicted 0s.

$1 - R^{(u)}(X)$ can be understood as a critical success index for rare events (Schaefer, 1990; Legrand et al., 2022), with asymptotic properties proven by Legrand et al. (2022), such as the link to the extremal index (we refer to their article for more details).

The risk function $R^{(u)}(X)$ is initially designed for deterministic forecasts. We extend it here to an index for ensemble forecasts, by comparing the observed exceedances with the median member of the forecast exceedances $F^{med}$. There are 11 members in the ECMWF precipitation hindcast data: for a given location, a given initialisation date and a lead time, $F^{med} = 1$ if at least 6 ensemble members predict extreme precipitation and $F^{med} = 0$ otherwise. We take here the median forecast across members, but in practice $F^{med} = 1$ could be set to 1 only if fewer or more than 6 members forecast extreme precipitation. The choice depends on the risk aversion of the users and is discussed in Section 4.

This adapted index is later on called the binary loss index ($BLI_m$, $m$ indicating the median of the ensemble members). It is defined by:

$$BLI_m = \frac{N_1^{med}}{N_2^{med}},$$

where $N_1^{med}$ is the number of days when the observation and the ensemble median disagree, i.e. $N_1^{med} = \#\{j \mid F_j^{med} \neq Y_j\}$, and $N_2^{med}$ is the number of days when an extreme event occurs in either or both the observation and the ensemble median, i.e. $N_2^{med} = \#\{j \mid \left(F_j^{med} = 1 \text{ or } Y_j = 1\right)\}$. In other words, $N_1^{med}$ is the number of false positives and false negatives and $N_2^{med}$ is the number of true positives, false positives and false negatives.

To measure the lead time dependence of the skill, $BLI$ is computed for each lead time day. Note that if the forecast $F$ and the observation $Y$ are independent (i.e. the forecast has no skill) and if $\mathbb{P}[F = 1] = \mathbb{P}[Y = 1] = \alpha$, then $BLI = \frac{2 - 2\alpha}{2 - \alpha}$ (here, $\alpha = 0.05$ for daily exceedances in a given season). In our case, $\mathbb{P}[F = 1]$ is not exactly equal to $\mathbb{P}[Y = 1]$ because the index is computed on extended seasons and not on 3-month seasons.

The climatological value of $BLI$, referred to as $BLI_{clim}$ is used as reference value. We compute confidence intervals for $BLI_{clim}$ with a bootstrap procedure to determine if the forecast is skillful, i.e. if $BLI$ is significantly lower than $BLI_{clim}$. For a given bootstrap step, a random time series is formed by drawing values in the observation time series. The $BLI$ is computed with this random time series as forecast. For a given lead time day, a forecast is deemed to be significantly skillful if the $BLI$ of the median member of the forecast ($F^{med}$) is lower than the 5% percentile of the confidence interval on $BLI_{clim}$. Like for the Brier score, we compute the "last skillful day" for the BLI, with the same definition (largest lead time day with skill, see figure 2 for an example).

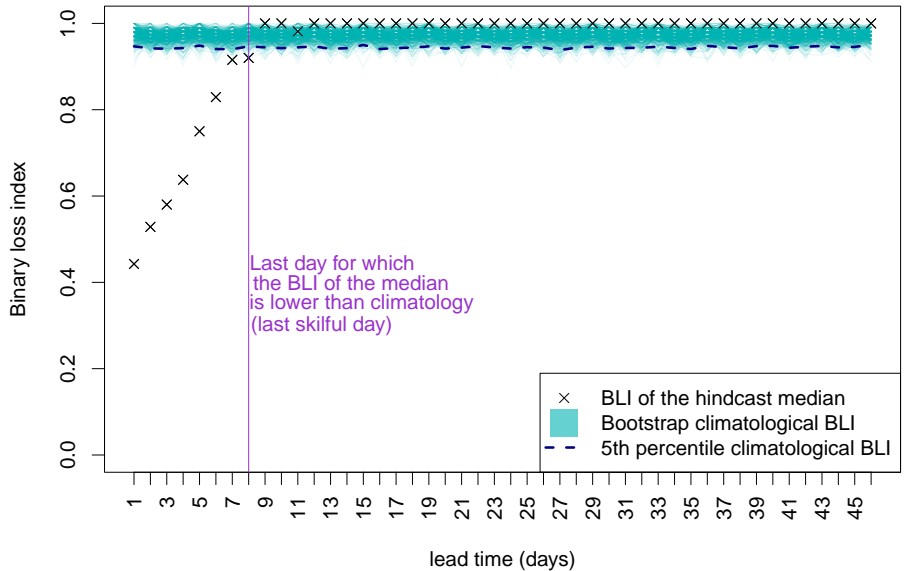

**Figure 2.** Definition of the last skillful day for BLI: example for one grid point and one season.

### 2.3.3 Spatio-temporal extension of the metrics

Requiring an exact match of events in the forecast and the observations on the same day and at the same grid point is very strict. Indeed, precipitation is a complex variable to forecast precisely in space and time. A forecast may contain useful information, even if the forecast does not predict the event exactly on the same day or at the same location as in the observation, but in a close neighborhood. Moreover, a temporal lag or a spatial shift between the observation and the forecast is penalized twice, by 1) a missed event at the observed time/location of the event and 2) a false alarm at the erroneously predicted time/location of the event (double penalty issue, see e.g. Ebert et al., 2013). We therefore also compute skill scores on data aggregated in space and time, which allows for some flexibility in the exact location or exact timing of the events. The spatial and the temporal aggregations are conducted independently, to analyse the individual impact of each aggregation. Both the spatial and temporal neighborhoods are non-overlapping to consider each extreme event only once. The spatio-temporal extensions are applied before computing the median member.

The temporal aggregation consists in counting the number of extreme events $N^t$ in a 7-day window. We then translate it into a binary series $E_n^t$: given a minimum number of events $n$ in the window ($n = \{1, ..., 7\}$) :

$$
E_n^t = \begin{cases} 1 & \text{if } N^t \geq n \\ 0 & \text{otherwise.} \end{cases}
$$

Figure 3 provides an example for the definition of $E_n^t$. To the various binary series (one for each $n$), we apply the Brier score and $BLI$ to quantify forecast skill. We estimate climatological skill in a way that conserves the temporal structure of the

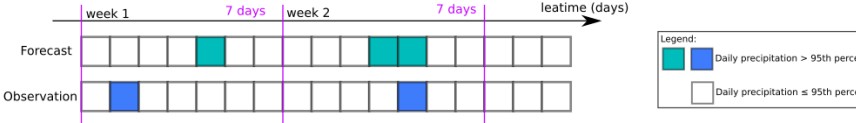

**Figure 3.** Illustration of the weekly aggregation of extremes at one grid point. During week 1, the forecast predicts one extreme and one extreme is observed. For both the forecast and the observation, the number of extreme events in the 7-day window is greater or equal to 1: $E_1^t = 1$ for the two datasets. For both datasets, the number of events in the 7-day window is lower than $n$, for $n \geq 2$: $E_n^t = 0$ for the two datasets. During week 2, one extreme is observed but the forecast predicts two events. For both datasets, the number of extreme events in the 7day window is greater or equal to 1: $E_1^t = 1$ for the two dataset. For the observation, the number of events in the 7-day window is lower than 2 ($E_2^t = 0$) and this number is greater or equal to 2 for the forecast ($E_2^t = 1$). For the configuration with $n$ events, $n \geq 3$, $E_n^t = 0$ for both datasets.

climatology. We randomly select the beginning of the 7-day time windows in the observation. The 6 following days are not randomly selected, they are the 6 days actually following the beginning of time window in the time series of observations.

The spatial aggregation is performed by counting extreme precipitation events in neighborhoods. Like for the temporal

aggregation, we define two categories, depending on whether the count of events $N^s$ in the spatial neighborhood exceeds or not some threshold $n$ (see figure 4 for an example):

$$E_n^s = \begin{cases} 1 & \text{if } N^s \geq n \\ 0 & \text{otherwise.} \end{cases}$$

Precipitation includes some spatial structure, i.e. spatial dependence between points in a neighborhood. When computing the climatology for both scores, the spatial structure is conserved: for one step of the bootstrap only the date is randomly chosen,

the spatial neighborhood is the observed neighborhood for that day. We define the neighborhoods as square boxes of about 150km*150km, i.e. boxes with a latitudinal extent of 1.5°N (3 gridboxes) and with a longitudinal grid extent that depends on the latitude: from 1.5°E at 30°N (3 gridboxes) to 4.5°E at 70°N (9 gridboxes), see figure D1 in appendix for an illustration.

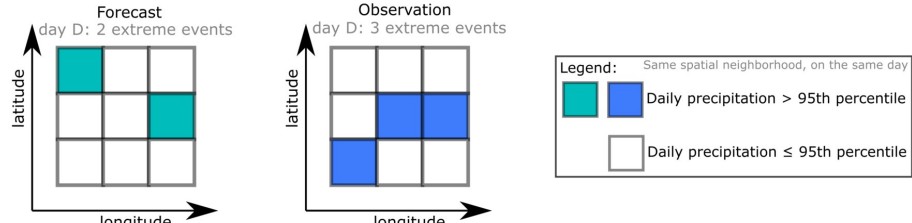

**Figure 4.** Illustration of the spatial aggregation of extremes in one neighborhood. The forecast indicates two extremes in the spatial neighborhood and three events are observed. For both datasets, the number of extreme events in the neighborhood is greater or equal to 1 ($E_1^s = 1$) and greater or equal to 2 ($E_2^s = 1$). For 3 events or more, $E_3^s = 0$ for the forecast and $E_3^s = 1$ for the observation. For four events or more, $E_n^s = 0$ for both datasets for $n \geq 4$.

## 3 Results

### 3.1 Daily and local comparison

We begin by discussing the forecast skill at the daily and grid-point scale. The BLI indicates more skill during the extended winter (skill for up to 11 days, and many regions with a last skillful day greater than 7 days) than during the extended summer (last skillful day below 6 days for most grid points), see figure 5. Regions with high skill are Norway, the Alps and the western half of the Iberian Peninsula in the extended winter and the Bay of Biscay, the South of France, Norway, Central Europe and the South of the Alps in the extended summer. The BLI skill score is less conservative than those of the Brier skill score, however 185 the spatial patterns are similar for the two metrics (figure B1 in the appendix). That is, the last skillful day for the Brier skill score is overall smaller than the last skillful day for the BLI, but both the Brier score and the BLI show the same regions with high and low skill of the forecast for precipitation extremes, in summer and winter.

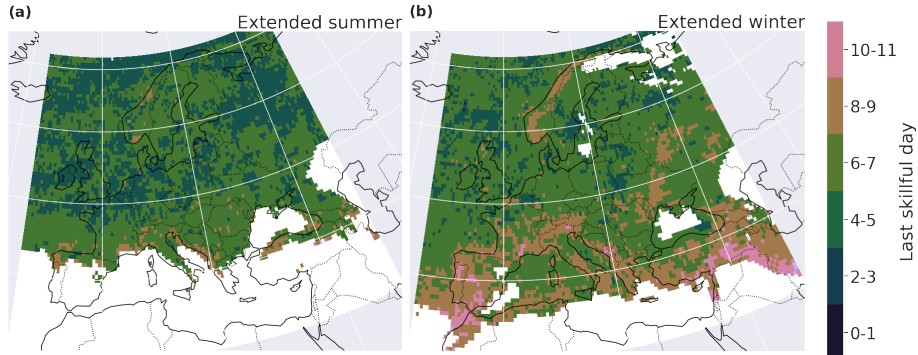

**Figure 5.** Last skillful day for the BLI for a local and daily comparison, in extended summer (a) and extended winter (b).

## 3.2 Temporal aggregation

7-day extreme precipitation event counts are also better predicted during the extended winter than during the extended summer (figure 6). For the category "one event or more occurred during the 7 days", the forecasts at most grid points still have skill for lead times into the second week, i.e. days 8 -14, in extended winter. The BLI decreases as the number of events per week increases; however, the spatial patterns remain the same. The regions where temporal clustering is more skillfully forecasted are the Iberian Peninsula, Norway and the northern Mediterranean coastline (especially in winter). The Brier score confirms these results, with similar patterns (see figure C1 in appendix).

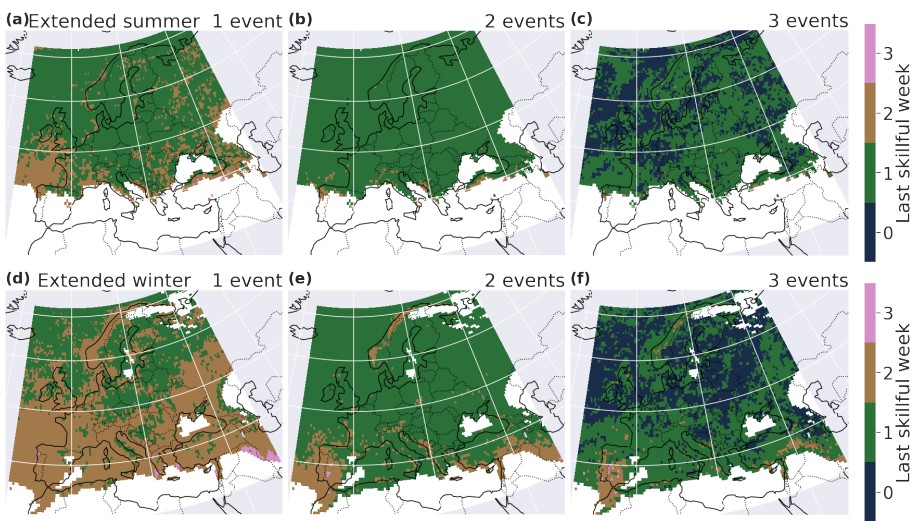

**Figure 6.** Last week of skill for the BLI in extended summer (a-c) and extended winter (d-e) for a minimum of 1 (a,d), 2 (b,e) and 3 (c,f) events in a 7-day window. A last skillful week equal to 0 means that, for the count of extremes during the first week lead time, the $BLI$ of the forecast is not significantly lower than $BLI_{clim}$.

## 3.3 Spatial aggregation

Extended winter forecasts for spatially aggregated extremes are also more skillful than summer ones (see figure 7). The last skillful day is greater when spatially aggregating that for the local analysis, but the two configurations have a similar spatial pattern. In extended winter, for one event or more in the neighborhood, the last skillful lead time reaches up to 11 days in many regions: the western Iberian Peninsula, the Norway coast, and the west-facing coasts in general. In extended summer, the last skillful lead time is between 8 and 11 days on the Atlantic coast of France, Italy, Western Europe and the coasts of the Iberian Peninsula . The spatial skill pattern remains similar with increasing number of events per neighborhood but the skill decreases.

Figure D2 in appendix shows maps of the last lead time day with a positive Brier skill score, for different numbers of events aggregated spatially, in extended summer and extended winter. The regions with higher skill are the same for the Brier score and for the BLI. The spatial pattern of the skill also remains similar with increasing number of events per neighborhood.

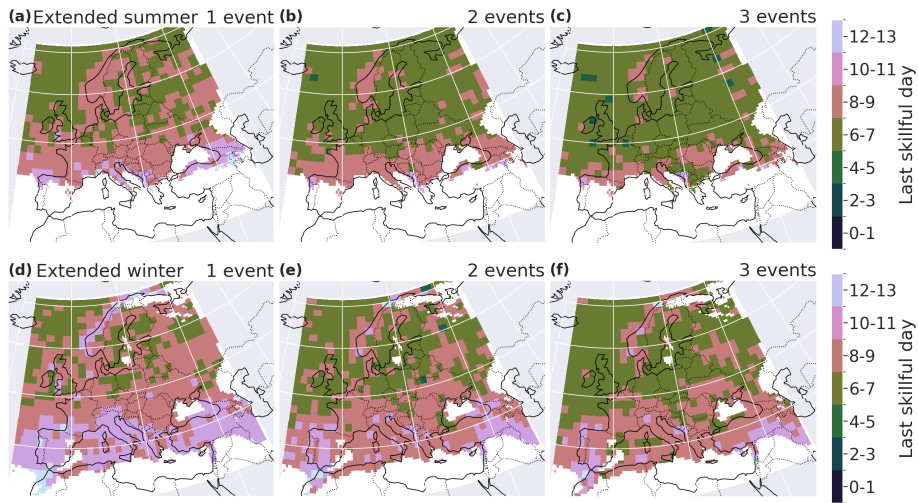

**Figure 7.** Last day of skill for the BLI in extended summer (a-c) and extended winter (d-f) for a minimum of 1 (first column), 2 (second column) and 3 (last column) events in neighborhoods of 150×150km.

## 4 Discussion and Conclusion

In this paper, we assess forecast skill of extreme precipitation occurrence over Europe in the ECMWF S2S model. Extremes are defined as exceedances over the seasonal $95^{th}$ percentile. We conduct a verification against ERA5 precipitation with the binary loss index ($BLI$) and the Brier score. We extend the binary loss score studied by Legrand et al. (2022), which was designed for deterministic forecasts only, to ensemble forecasts. We define the $BLI$ as the binary loss score calculated for the ensemble median member of binary exceedances. The choice of the median member was motivated by a trade-off between false alarms and missed events. The skill will be different when one chooses a lower percentile of the ensemble members to compute the $BLI$ (risk averse setting) or when one chooses a high percentile of the ensemble members (risk loving). The $BLI$ has the advantage of focusing exclusively on extreme event occurrence (hit, false alarm or miss) and is not biased by the high counts of extreme event non-occurrence. The $BLI$ is qualitatively compared with the Brier score; the skill scores of two metrics agree very well over Europe. Despite the great importance of accurately forecasting rare extremes, the Brier score does not give a special weight to underrepresented classes. Therefore, by design, the $BLI$ should be preferred to the Brier score when assessing the forecast skill for very rare events. With further research, a probability score for ensemble forecasts could be developed from the $BLI$.

The S2S forecasts have overall higher skill in predicting extreme precipitation events in winter than in summer. A likely explanation resides in the fact that precipitation over Europe mainly results from large scale processes during winter, but from small scale, convective events in summer. Predicting small scale events is indeed more challenging than large scale ones (Haylock and Goodess, 2004; Kenyon and Hegerl, 2010). This result is in agreement with the existing literature on S2S prediction in other regions (Tian et al., 2017; Kolachian and Saghafian, 2019). Norway, Portugal and the South of the Alps

are regions with the most skill. The orography seems to be a source of skill (like in Norway, the Pyrenees and the South of the Alps): the forecast seems to better capture precipitation events where the complex topography acts as a forcing for precipitation. The Mediterranean region exhibits relatively good skill in winter. Similarly, coastal regions in general have a higher skill compared to continental regions. A potential explanation for this difference is that the water transported from the ocean first rains out next to the coast; it is more challenging to predict where the remaining water in the atmosphere will rain down on continental regions because land-atmosphere interactions introduce uncertainty. A follow-up study could further investigate these hypotheses on the physical reasons behind the spatial and seasonal heterogeneity of the skill.

Allowing for temporal or spatial flexibility in the evaluation of the forecast extremes confirms the skill patterns, bringing robustness to the analysis. The skill for the spatially aggregated precipitation is slightly larger than for the local analysis, as it is easier for the forecast to have a matching event with observation on a larger grid. The spatial aggregation conducted here could be adapted for an impact-oriented analysis, by aggregating e.g. over catchments to evaluate the predictability of heavy precipitation, that can potentially result in floods or by analysing multi-day heatwaves.

We additionally investigated the effect of European weather regimes on the forecast skill (as defined in Grams et al., 2017), as the forecast skill of the weather regimes themselves can largely differ (Büeler et al., 2021). We computed the forecast skill independently for positive phases and negative phases of the NAO. The forecast skill does not exhibit a strong dependence on the NAO phase, although the data was also spatially aggregated to increase robustness (not shown). This absence of signal should be confirmed with a deeper analysis, by considering some time lag or seasonality for the influence of the teleconnection patterns (Tabari and Willems, 2018) or by aggregating over larger spatio-temporal neighborhoods, to increase the robustness. Other teleconnection patterns could be investigated, such as Scandinavian and East Atlantic patterns, El Niño southern oscillation, the Atlantic multidecadal oscillation (Casanueva et al., 2014) or the state of the stratosphere (Domeisen et al., 2019).

An assessment focused on the precipitation intensity could extend the verification; the precipitation forecast data would then require to be calibrated (Gneiting et al., 2007; Specq and Batté, 2020; Crochemore et al., 2016; Monhart et al., 2018; Huang et al., 2022). An extension of the CRPS would be an option to measure the intensity forecast skill with a focus on heavy precipitation, like the threshold-weighted CRPS (see e.g. Pantillon et al., 2018; Allen et al., 2021) or using extreme value theory (Taillardat et al., 2022). Post-processing the hindcast data and analysing the paradigm of "maximizing the sharpness of the predictive distributions subject to calibration" could also be an extension of this work (Gneiting et al., 2007); the usual evaluation metrics –the probability integral transform histogram, marginal calibration plots, the sharpness diagram– could be applied with a focus on extremes.

Note that for practical applications, one needs caution to interpret the skill in an absolute way, for two reasons: (i) a skillful forecast does not mean that the forecast is also useful forecast for practical applications and (ii) the absolute last skillful day depends on the choice of the member for the daily predictor (here, the median member). (i) If the BLI is equal to 0.8 but is outside of the climatological confidence interval, the forecast is better than the climatology and therefore skillful. However, it also means that only 25% of the extremes are caught by the forecast (by simple transformation of $BLI = \frac{FN+FP}{TP+FN+FP} = 0.8$, where FN are the false negatives, FP are the false positives and TP are the true positives). 75% of the time, either the forecast erroneously predicted an extreme (false alarm, FP) or did not predict an extreme that occurred (miss, FN). The definition of

the last skillful day can be adapted depending on the usage of the forecast. The definition can be more conservative, e.g. the last lead time day for which at least 75% of the extreme events are caught (rather than a comparison to the climatology), or using a smaller percentile of the members, rather than the median member. (ii) The last skillful day is larger when choosing the maximum member as the daily predictor (i.e. $F^{med} = 1$ if at least 1 ensemble member predicts extreme precipitation and $F^{med} = 0$ otherwise). This is due to the number of TP not collapsing to zero with increasing lead times: the condition "at least one member predicts daily precipitation is greater than the $95^{th}$ percentile" is very easily satisfied. By "chance", the maximum member predicts many TP, still compensating a bit for the FP for large lead times. For the choice of the member, we recommend considering a good balance between FN and FP. However, it is important to note that the spatial pattern of skill does not depend on the choice of the member. The regions with a relatively larger skill (e.g. Norway, Portugal, West coasts in winter) remain the same, independently of the choice of predictor (minimum, median or maximum member). Following these two remarks, we emphasize that our aim here was to provide a robust qualitative assessment, by identifying regions of skill and challenging regions for the forecast model to predict precipitation extremes on the S2S timescale.

Checking if a value of the BLI is significant is a kind of hypothesis test that is repeated for a large number of grid points. One could argue that some regional significance should be investigated. However, when displaying the local significance as "largest lead time day with skillful forecast", the results are continuous rather than a strict "yes or no" response. Moreover, the spatial coherence of the results confirms the robustness of the method.

Our method to assess extremes can also be applied to other variables, such as consecutive days of high temperature, river discharge, etc. Considering the other side of extremes, evaluating the skill of forecasts to predict droughts would also be of crucial importance. For droughts, the persistence of dry periods matters, rather than the occurrence of precipitation. The method could be adapted accordingly, e.g. adjusting the definition temporal aggregation introduced in this study.

*Code and data availability.* The codes used for the data analysis are available on github (https://github.com/PauRiv/S2S_verif_precip).
The ECMWF's S2S hindcast data are available on the ECMWF platform
(https://apps.ecmwf.int/datasets/data/s2s-reforecasts-instantaneous-accum-ecmf/levtype=sfc/type=cf/, cycle 47r2).

# Appendix A: $95^{th}$ percentile

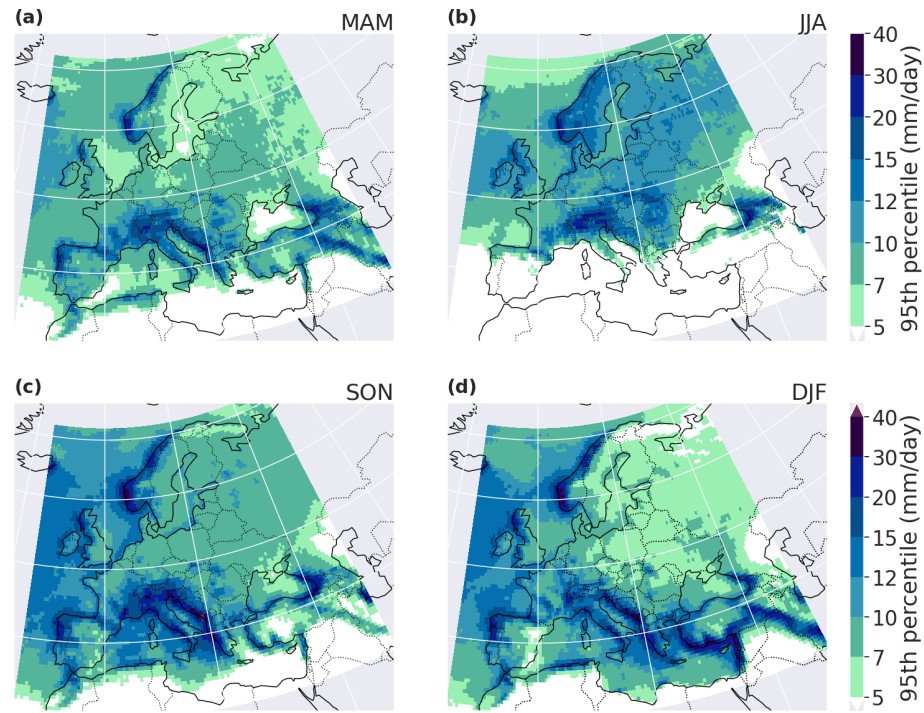

**Figure A1.** 95$^{th}$ percentile of daily precipitation in ERA5, 2001-2021.

## Appendix B: Local and daily comparison of extremes

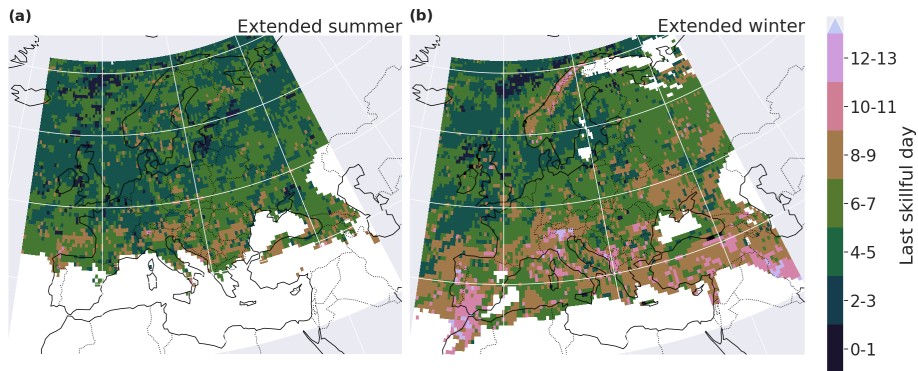

**Figure B1.** Last skillful day for the Brier skill score for local and daily comparison, in extended summer (a) and extended winter (b).

## Appendix C: Temporally accumulated extremes

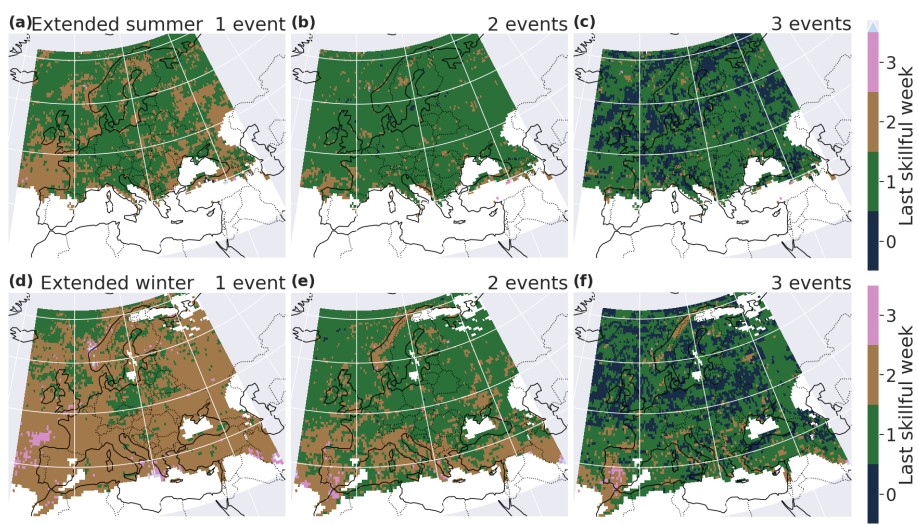

**Figure C1.** Last week of skill for the Brier skill score in extended summer (a-c) and extended winter (d-f) for a minimum of 1 (first column), 2 (second column) and 3 (last column) events in a 7 days window.

## 285 Appendix D: Spatially accumulated extremes

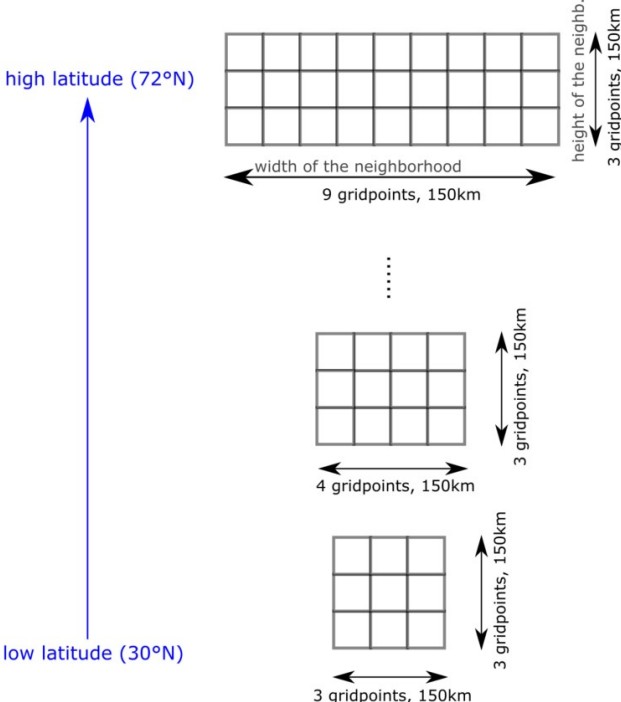

**Figure D1.** Illustration of the width of the spatial neighborhood, in terms of grid points, depending on the latitude for a constant width in kilometers (and for a constant area).

*Author contributions.* PR, OM and PN designed the research. PR executed the analysis, created the figures and wrote the draft. OM and PN contributed to the formal analysis. OM, PN and AT contributed to the data analysis and revised the manuscript. A.T. collected the data. All authors contributed to the final version of the manuscript.

*Competing interests.* The authors declare that they have no conflict of interest.

*Acknowledgements.* P.R. and O.M. acknowledge funding from the Swiss National Science Foundation (grant number 178751). We thank Jonas Bhend for helpful discussions. Part of P.N. work was supported by three French national programs (80 PRIME CNRS-INSU, ANR T-REX under reference ANR-20-CE40-0025-01, ANR Melody (ANR-19-CE46-0011), and the European H2020 XAIDA (Grant agreement ID: 101003469). The support of DAMOCLES-COST-ACTION on compound events is also acknowledged.

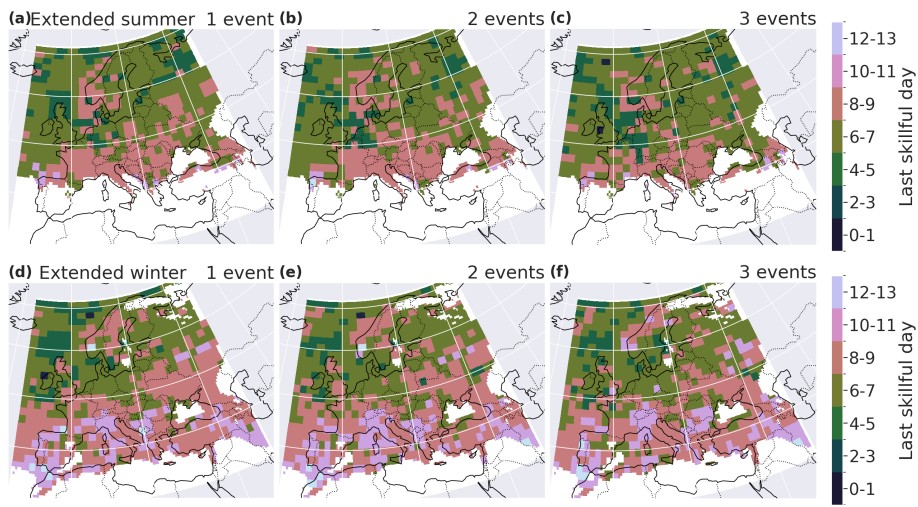

**Figure D2.** Last day of skill for the Brier skill score in extended summer (a-c) and extended winter (d-f) for a minimum of 1 (first column), 2 (second column) and 3 (last column) events in neighborhoods of 150×150km.

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
