# Peer review of "Assessment of S2S ensemble extreme precipitation forecast skill over Europe"

_EGUsphere, 2022_

## Referee Comment (RC1)

**Review of 'Assessment of S2S ensemble extreme precipitation forecasts over Europe'**

In this paper, the authors characterise the forecast skill of extreme precipitation in the ECMWF S2S hindcast ensemble, taking some account of seasonality, and considering the impacts of spatial and temporal aggregation on the forecast skill. They measure skill in a deterministic way using the Brier score and using the binary loss index.

They find that extended winter is more predictable than extended summer, and that temporal and spatial aggregation both extend the last skilful day. They also identify several regions where forecast skill is higher: Norway, Western Iberia and the South of France. In general more mountainous and coastal areas show higher skill.

I like the analysis approach, and the temporal and spatial aggregation is well done, as is the bootstrap approach to determining the last skilful day. I have a minor methodological question, but I also see some more substantial issues and so am recommending major revisions.

Firstly, I am a little confused by the choice of the BLI as a metric. A fair amount of time is spent discussing this new metric, but as the authors point out it is not at all novel in meteorology, simply being (1 - the well known Critical Success Index). This raises the question of why not just use the more established CSI from the beginning?

More generally, my main sense is that the analysis, while well done, is quite basic. Skill maps are computed for some variant event definitions using two different seasons and scores, and we are done. As you discuss in your conclusions, there are many interesting questions that arise from this foundation. I think the manuscript could do with answering at least some of them.

Without suggesting you wildly broaden the scope, I suggest answering the following questions:
- How does the skilful day change for different cost-loss ratios (i.e. using different ensemble thresholds)? Understanding this sensitivity has a lot of real-world relevance. To simplify presentation you could average skill over a few boxes of interest, so you had scalar values for each threshold.
- How does the spatial pattern of skill change for compound events? Yes, we can in theory read this from figures 6 and 7, but perhaps some anomaly plots might be helpful here.

These are only ideas; the main point for me is that some extra richness is needed, whether that be a discussion of regional differences, dynamical drivers, sensitivity of the results, a deeper analysis of scale-dependence etc.

On top of that the current comparison of the Brier score and BLI are a bit superficial,

and should be discussed in more detail.

**Minor comments**
- The Github repository, which is supposed to contain the code, is empty
- You define the forecast event thresholds using the forecast data, conditional on both season and lead time, rather than using the observational thresholds to account for bias. But is there a risk here that you make the model seem too good? You are bias correcting with your testing data! If you were to set thresholds with only half your data and then test the skill in the other half, would the skill go down? I would like to see this for at least the unaggregated case.
- Can you make sure figures 5, 7, B1 and D2 all use the same colorbar? Currently it is hard to compare them.

**Very Minor comments**

- References need formatting to be in parentheses
- 'Skilfull' → 'skilful' L185
- 'BLF' → 'BLI' L204

---

## Author Comment (AC1)

**Answer to Reviewer n.1**

We would like to thank the reviewer for taking the time to review and provide helpful comments to improve our manuscript. In the following, we answer (in blue) to the points raised by the reviewer (in black), and we indicate how we adapted the manuscript (in green).
* * *
Review of 'Assessment of S2S ensemble extreme precipitation forecasts over Europe'

In this paper, the authors characterise the forecast skill of extreme precipitation in the ECMWF S2S hindcast ensemble, taking some account of seasonality, and considering the impacts of spatial and temporal aggregation on the forecast skill. They measure skill in a deterministic way using the Brier score and using the binary loss index.

They find that extended winter is more predictable than extended summer, and that temporal and spatial aggregation both extend the last skilful day. They also identify several regions where forecast skill is higher: Norway, Western Iberia and the South of France. In general more mountainous and coastal areas show higher skill. I like the analysis approach, and the temporal and spatial aggregation is well done, as is the bootstrap approach to determining the last skilful day. I have a minor methodological question, but I also see some more substantial issues and so am recommending major revisions.

Firstly, I am a little confused by the choice of the BLI as a metric. A fair amount of time is spent discussing this new metric, but as the authors point out it is not at all novel in meteorology, simply being (1 - the well known Critical Success Index). This raises the question of why not just use the more established CSI from the beginning?
* * *
Thank you very much for this comment. The BLI can indeed be seen as "1 - the well known CSI", for rare events. As Legrand et al. (2021) focused on the negatively oriented risk function, we kept this configuration. The results would be symmetrically similar with the index 1-BLI. We would like to mention that we use the "last skilfull day" as a skill measure. The CSI does not itself provide any information on the skill.

Following your question, we extended the paragraph introducing the BLI and further improved further the justification of its use:

"Legrand et al. (2021) studied in detail a risk function defined as the ratio between the empirical probability of having an extreme event in either the observation dataset or the forecast dataset, and the empirical probability of having an extreme event in the observations or the forecast (including having an event in both datasets). In our context, the risk function can be written:

$$R^{(u)}(X)=\frac{\mathbb{P}(X^{(u)} \neq Y^{(u)}) }{\mathbb{P}(Y^{(u)}=1 \textrm{ or } X^{(u)}=1 )}$$

where $Y^{(u)}$ is the binary observation, $Y^{(u)}=0$ (resp. $Y^{(u)}=1$) if the observed daily precipitation is lower (greater) than a certain threshold u; and $X^{(u)}$ is the binary forecast $X^{(u)}=0$ (resp. $X^{(u)}=1$) if the predicted daily precipitation is lower (greater) than u.

The risk function $R^{(u)}$ focuses on how well the "1" values (extreme event days) match between observation and forecast. It does not take into account steps when neither the forecast nor the observation experience an extreme event. $R^{(u)}(X)$ varies between [0;1] and is negatively oriented (the closer to zero, the better the forecast is). The strength of $R^{(u)}(X)$ is its asymptotic behavior: even for very rare events, both the over-optimistic and over-pessimistic forecasts will be penalized. In other words, even for very large threshold u, i.e. *Y=1* for very rare occasions (but at least once), if the forecast is too optimistic and *X=0* for all time steps, then $R^{(u)}(X)=1$ ("naive" classifier, Legrand et al., 2021). A very pessimistic forecast will be penalized the same way ("crying-wolf" classifier, see Legrand et al., 2021). The Brier score rather assesses the average behavior, with a very weak penalty for under-represented classes. Because all days are compared, the assessment of rare extreme events (missed, false alarm or hit) by the Brier score is lost among the huge amount of correctly predicted 0s.

*$1-R^{(u)}(X)$* can be understood as a critical success index for rare events (Schaefer, 1990; Legrand et al., 2021), with asymptotic properties proven by Legrand et al. (2021), such as the link to the extremal index (we refer to their article for more details)."
* * *
More generally, my main sense is that the analysis, while well done, is quite basic. Skill maps are computed for some variant event definitions using two different seasons and scores, and we are done. As you discuss in your conclusions, there are many interesting questions that arise from this foundation. I think the manuscript could do with answering at least some of them.
Without suggesting you wildly broaden the scope, I suggest answering the following questions:

• How does the skilful day change for different cost-loss ratios (i.e. using different ensemble thresholds)?
Understanding this sensitivity has a lot of real-world relevance. To simplify presentation you could average skill over a few boxes of interest, so you had scalar values for each threshold.
* * *
Thank you for raising this point. It is indeed interesting to conduct a sensitivity analysis on the cost-loss ratio. The choice for the ensemble threshold is basically a tradeoff between missed events or false alarms.
To develop this topic, we compared the last skillful day when using
-the minimum member as predictor, i.e. $F^{med}=1$ if all the 11 ensemble members predict extreme precipitation and $F^{med}=0$ otherwise (Fig. 1_r1);
-the maximum member as predictor, i.e. $F^{med}=1$ if at least 1 ensemble member predicts extreme precipitation and $F^{med}=0$ otherwise (Fig. 2_r1).

The two important points brought by Figures 1_r1 and 2_r1 are:
   1) The last skillful day is increasing with a less conservative definition of the predictor. There is a consistent signal over Europe : lower last skillful day for the minimum member, intermediate last skillful day for the median (figure 5 in the paper), and greater
   2) The spatial pattern is robust to this sensitivity analysis. The regions with a relative larger skill (e.g. Norway, Portugal, West coasts in winter) are the same, independently of the choice of predictor (minimum, median or maximum member)
Point 2) is crucial for our analysis: we aim to provide a qualitative assessment, identifying regions of skill and regions where the forecast is facing challenges to predict precipitation extremes on the S2S timescale.
In the next lines, we develop point 1)

[Figure]

*Figure 1_r1: Last day of skill for the BLI in summer (a) and winter (b) taking the minimum member as predictor (for each day and each initialisation date).*

[Figure]

*Figure 2_r1: Last day of skill for the BLI in summer (a) and winter (b) taking the minimum member as predictor (for each day and each initialisation date).*

To understand the quantitative dependance of the last skillful day on the chosen member, we take the example of a grid point in France (near South West of Lyon, North of Saint Etienne, longitude 4.35°W latitude 45.65°N), in extended winter. The line of reasoning is the same for other grid points and summer.

The differences in last skillful day seems to be due to the asymptotic value of the BLI : when choosing the maximum member (or the 8,9,10th member), the BLI does not converge to 1, even for large lead times (Fig. 3_r1). Additionally, one can note that the median member has one of the lowest BLI for short leadtimes, suggesting an excellent compromise between false negatives and false positives.

[Figure]

*Figure 3_r1: Evolution of the BLI with lead time, depending on the member chosen as predictor (color). This figure is similar to Fig. 2 in the manuscript. Grid point: longitude 4.35°W latitude 45.65°N, in extended winter.*

We recall the definition of the BLI:

$$BLI = \frac{FP + FN}{TP + FP + FN}$$

The BLI converges to 1 when the number of true positives (TP) becomes negligible compared to the false negatives (FN) and false positives (FP). It is not the case for the maximum member, as shown in Figure 4_r1. The true positive rate (TPR) does not collapse to zero with large lead times.

[Figure]

*Figure 4_r1: Same as Fig. 3_r1 but for True Positive Rate (TPR).*

[Figure]

*Figure 5_r1:* Case study of an extreme precipitation event and its forecast for different predictors (minimum, median or maximum member), over Switzerland (2002-06-05). Blue indicates whether precipitation was above the 95th percentile, for both observation and hindcast. Green color indicates a hit (true positive) by the forecast, yellow a false positive (false alarm) and purple a false negative (missed event)

With the case study of an extreme precipitation event over Switzerland in summer, Figure 5_r1 illustrates the fact that TPR is not converging to zero for the maximum member (same line of reasoning for other regions and winter). The 95th percentile is not "extreme enough" for the BLI to adopt the asymptotic behavior of the risk function introduced by Legrand et al (2022), preventing the "crying-wolf" situation. The condition "at least one member predicts daily precipitation > 95th" is very easily satisfied, so the maximum member predictor predicts an extreme event for many gridpoints. By "chance", the maximum member predicts a lot of true positives, much more than the median or minimum predictors for large lead times. But it also creates a lot of false alarms.
We therefore recommend the user to consider the balance between false negatives and false positives when choosing the member being the predictor: artifacts can appear for the least conservative definitions.

In the discussion section, we extended the paragraph about the interpretation of the last skillful day:

"Note that for practical applications, one needs caution to interpret the skill in an absolute way, for two reasons: (i) a skillful forecast does not mean that the forecast is also useful forecast for practical applications and (ii) the absolute last skillful day depends on the choice of the member for the daily predictor (here, the median member). (i) If the BLI is equal to 0.8 but is outside of the climatological confidence interval, the forecast is better than the climatology and therefore skillful. However, it also means that only 25\% of the extremes are caught by the forecast (by simple transformation of $BLI=\frac{FN + FP}{TP + FN + FP}=0.8$, where FN are the false negatives, FP are the false positives and TP are the true positives). 75\% of the time, either the forecast erroneously predicted an extreme (false alarm, FP) or did not predict an extreme that occurred (miss, FN). The definition of the last skillful day can be adapted depending on the usage of the forecast. The definition can be more conservative, e.g. the last lead time day for which at least 75\% of the extreme events are caught (rather than a comparison to the climatology), or using a smaller percentile of the members, rather than the median member. (ii) The last skillful day is larger when choosing the maximum member as the daily predictor (i.e. $F^{med}=1$ if at least 1 ensemble member predicts extreme precipitation and $F^{med}=0$ otherwise). This is due to the number of TP not collapsing to zero with increasing lead times: the condition ``at least one member predicts daily precipitation is greater than the 95th percentile'' is very easily satisfied. By ``chance'', the maximum member predicts many TP, still compensating a bit for the FP for large lead times. For the choice of the member, we recommend considering a good balance between FN and FP. However, it is important to note that the spatial pattern of skill does not depend on the choice of the member. The regions with a relatively larger skill (e.g. Norway, Portugal, West coasts in winter) remain the same, independently of the choice of predictor (minimum, median or maximum member). Following these two remarks, we emphasize that our aim here was to provide a robust qualitative assessment, by identifying regions of skill and challenging regions for the forecast model to predict precipitation extremes on the S2S timescale."

We also added slight modifications in the method section of the BLI, for clarity:
- we added ", a given initialisation date and a lead time" in the sentence "There are 11 members in the ECMWF precipitation hindcast data: for a given location, a given initialisation date and a lead time, $F^{med}=1$ if at least 6 ensemble members predict extreme precipitation and $F^{med}=0$ otherwise.". We want to make sure the reader understands that the ensemble member chosen, the median member, can be a different ECMWF member everyday.
- we added " and is discussed in Section 4" at the end of the same paragraph

• How does the spatial pattern of skill change for compound events? Yes, we can in theory read this from figures 6 and 7, but perhaps some anomaly plots might be helpful here.
* * *
Thank you for the suggestion. It is indeed easier to compare with the same colorbar. We can see that the skill is generally increased by aggregating spatially.

We added a sentence in the result section (" The last skillful day is greater when spatially aggregating that for the local analysis, but the two configurations have a similar spatial pattern.") and in the discussion (" The skill for the spatially aggregated precipitation is slightly larger than for the local analysis, as it is easier for the forecast to have a matching event with observation on a larger grid.") .

Fig 6_r1 confirms this general increase of skill, and does not highlight any significant pattern for the magnitude of change.

[Figure]

*Fig 6_r1: Difference of last skillful day between spatially aggregated precipitation and local precipitation*
* * *
These are only ideas; the main point for me is that some extra richness is needed, whether that be a discussion of regional differences, dynamical drivers, sensitivity of the results, a deeper analysis of scale-dependence etc.
* * *
Thank you for these suggestions.

With the new colormaps, it is now easier to compare the skill on the spatio-temporal scale (increase of skill).

In response to Reviewer n.2 pointing to a deeper analysis of the dynamical drivers, we added in the discussion part a paragraph on our analysis of the dependance of the skill on the NAO regime. We would also like to specify that our analysis aims to provide an overall assessment of the forecast skill, using a metric robust to the rarity of events. We include in the discussion section the hypothetical reasons for temporal and spatial heterogeneities of the skill (e.g. moisture origin and orography). We added the sentence "A follow-up study could further investigate these hypotheses on the physical reasons behind the spatial and seasonal heterogeneity of the skill."
* * *
On top of that the current comparison of the Brier score and BLI are a bit superficial, and should be discussed in more detail.
* * *
Thank you for your remark.

The point we wanted to highlight by comparing the BLI with the Brier score, is that they highlight the same regions with high and limited skill. We can see that the difference between the last skillful day of the two metrics is rather noisy (fig 7_r1), with no obvious pattern. The histogram of the difference (fig 8_r1) confirms that the difference is low, mainly greater than -1 day and lower than 2 days. The difference is rather positive, indicating that the BLI is slightly less conservative.

[Figure]

Fig 7_r1: Difference of last skillful day between BLI and BSS

fig 8_r1: Difference of last skillful day between BLI and BSS
(histogram for all gridpoints)

We decided to not add these figures, as they do not bring a lot to the discussion. In the revised manuscript, we extended the comparison between the Brier score and the BLI, in the result section
" The BLI skill score is less conservative than those of the Brier skill score, however the spatial patterns are similar for the two metrics (figure \ref{fig:BSS_daily} in the appendix). In other words, the last skillful day for the Brier skill score is overall smaller than the last skillful day for the BLI, but both the Brier score and the BLI show the same regions with high and low skill of the forecast for precipitation extremes, in summer and winter."
Moreover, we added a sentence in the result part, to justify the choice of the metric:
" Despite the great importance of accurately forecasting rare extremes, the Brier score does not give a special weight to underrepresented classes. Therefore, by design, the BLI should be preferred to the Brier score when assessing the forecast skill for very rare events."
* * *
Minor comments
• The Github repository, which is supposed to contain the code, is empty
* * *
We apologize for this oversight. The codes to compute the BLI and the last skillful index, and to plot the figures have been uploaded.

• You define the forecast event thresholds using the forecast data, conditional on both season and lead time, rather than using the observational thresholds to account for bias. But is there a risk here that you make the model seem too good? You are bias correcting with your testing data! If you were to set thresholds with only half your data and then test the skill in the other half, would the skill go down? I would like to see this for at least the unaggregated case.

Thank you for this remark. Indeed we corrected the hindcast intensity bias with the hindcast climatology. The idea behind is that we want to focus on the assessment of extreme occurrence only, not intensity. In other words, we want to assess whether the 5% most extreme precipitation events in the hindcast are matching in time with the 5% observed ones. Hindcasts are created to obtain more data for the assessment of the "climatological" behavior of the forecast model, and can be used to correct the bias of the operational forecast. For the sake of simplicity, we used all 20 years to compute the 95th percentile. Removing one year (the year of the day to be corrected) would not have a strong impact, as all the years are run with the same forecast version, hence the same distribution.

• Can you make sure figures 5, 7, B1 and D2 all use the same colorbar?
Currently it is hard to compare them.

Thank you for this helpful remark, we modified the figures accordingly.

Very Minor comments
• References need formatting to be in parentheses
• 'Skilfull'→'skilful' L185
• 'BLF'→ 'BLI' L204

Thank you for pointing out these typos and mistakes, we modified the manuscript accordingly.

---

## Author Comment (AC2)

**Answer to Reviewer n.2**

We would like to thank the reviewer for their time and valuable feedback to improve our manuscript. In the following, we answer (in blue) to the points raised by the reviewer (in black), and we indicate how we adapted the manuscript (in green).
* * *
The authors aim to investigate the predictive ability of ECMWF S2S reforecast, focusing on intense precipitation events. S2S hindcast cover a range between medium-range and seasonal prediction. Currently this is one of the first work covering the skill of predicting extreme events at these long time ranges, is therefore interesting. They propose a relatively new index in measuring forecast accuracy (Binary Loss Index) although probably the same consideration would have been emerged using the Critical Success Index, the latter widely used in the meteorological community.

While the index computations and results are well presented, it is also true that discussion is weak in terms of meteorological implications. A part the obvious dependence of seasonality, with convective precipitation leading to less skill during summer, and spatial and temporal aggregation, would have been nice to investigate with greater detail the spatial variability of the lead time. I found the lead time definition useful and interesting, not as a number per se, as you correctly comment it depends on the level of event detection you want to achieve and it is user dependent, but as an index to investigate predictability and its dependency on other factors. Taking two regions of example, would have been interesting to composite days with very long time day, and events with a shorter long time day and show the differences in some upper level variable to infer the role of the precursors dynamical evolution leading to the precipitation extreme. In a way this goal was also mentioned in the introduction "*Skill information is also useful to identify potential sources of predictability and windows of opportunity (i.e. intermittent time periods with higher skill Mariotti et al. (2020))*". The long time day could be used to detect those windows of opportunity; in which conditions they occur and for which regions are stronger. In that respect I think some elements in the paper could be inserted.
* * *
Thank you for this remark, suggesting clarifying the goals of our manuscript. Our aim is to quantify the skill of weather forecast in predicting precipitation extremes on a S2S timescale. We mentioned the concept of windows of opportunities in the introduction as a motivation, as it is opening a whole new research question beyond the scope of our manuscript. Our analysis paves the way for deeper studies of the skill limitation and potential opportunities. Hypotheses regarding the physical reasons (such as moisture origin and orography) behind the spatial and seasonal heterogeneity of the skill are formulated in the discussion section, in the second paragraph. To improve the manuscript thank to the reviewer's remark, we modified the manuscript by:

a) adding the word "skill" in the title: "Assessment of S2S ensemble extreme precipitation forecast skill over Europe". The title is now more precise regarding the content of the manuscript, which is centered on the skill quantification of extreme precipitation forecast (with the binary loss index and the last skilful day), rather than the analysis of the processes responsible for skill or absence of skill in forecasting precipitation extremes.

b) extending the discussion section: we added a paragraph about the (absence of significant) results regarding the dependance of the skill on the NAO configuration.

"We additionally investigated the effect of European weather regimes on the forecast skill (as defined in Grams et al., 2017), as the forecast skill of the weather regimes themselves can largely differ (Büeler et al., 2021). We computed the forecast skill independently for positive phases and negative phases of the NAO. The forecast skill does not exhibit a strong dependence on the NAO phase, although the data was also spatially aggregated to increase robustness (not shown). This absence of signal should be confirmed with a deeper analysis, by considering some time lag or seasonality for the influence of the teleconnection patterns (Tabari and Willems, 2018) or by aggregating over larger spatio-temporal neighborhoods, to increase the robustness. Other teleconnection patterns could be investigated, such as Scandinavian and East Atlantic patterns, El Niño southern oscillation, the Atlantic multidecadal oscillation (Casanueva et al., 2014) or the state of the stratosphere (Domeisen et al., 2019)."

Figures 1_r2 and 2_r2 illustrate the limited signal obtained when computing the BLI on the different NAO regimes.The results for the Brier skill score are similar.

[Figure]

*Figure 1_r2: Last day of skill for the BLI in positive NAO phase (a-c) and negative NAO phase (d-f) for a minimum of 1 (first column), 2 (second column) and 3 (last column) events in neighborhoods of 150x150km.*

Forecasts for northern Norway have a higher skill during positive NAO phases than during negative phases, and forecasts for southern Europe have higher skill in negative phases of the NAO (Figure 1.). Apart from these regions, no clear patterns appear and the results are noisier than for the seasonal analysis. In general, in the negative phase of the NAO there is higher skill than in the positive phase and in the positive phase there is a larger latitudinal gradient of skill. However these results are not robust (Fig 2_r2.). The zonal mean of the skill length does not have a stronger gradient in one of the NAO phases when compared to the skill in the extended seasons. 20 years of hindcast data seem to not include a sufficient number of extreme precipitation days to distinguish the skill between the different NAO phases with our method.

[Figure]

*Figure 2_r2: Zonal mean of the last skilful day for the BLF, during the different NAO phases, and during extended summer and extended winter, for comparison (with spatial aggregation). The smaller last skillful day for NAO phases compared to summer and winter is due to shorter time series.*
* * *
I have mixed feelings on the final judgment of the work. Since the title is Assessment of S2S ensemble extreme precipitation forecasts over Europe, I was expecting a more in depth discussion on practical predictability limit of precipitation extremes. For this reason I finally opted for major revision because I think the material is insufficient for this topic. But if you just wanted to present a new method to score extreme forecast, as you say in the abstract *" The goal of this article is to introduce a new methodology to assess the skill of rare events"*, then the material could be sufficient but text and title needs to be restructured to put the accent on method. In the latter case a more in depth comparisons with results obtained with other scores is also needed.
* * *
Thank you for this insightful comment. As specified above, we slightly modified the title, adding of the word "skill" for more precision. We also modified the abstract, refining the goal formulation: "The goal of this article is to assess the forecast skill of rare events, here extreme precipitation, in S2S forecasts, using a metric specifically designed for extremes".

Additionally, following the remark from Reviewer n.1, we justified further the use of the Binary loss index, anchored in extreme value theory (see the new version of the method section "2.3.2 Binary loss index"). The comparison with the commonly used Brier score serves as a reference. We added a sentence about the limits of the Brier score in the method section 2.3.2 ("The commonly-used Brier score rather assesses the average behavior, with a very weak penalty for under-represented classes. Because all

days are compared, the assessment of rare extreme events (missed, false alarm or hit) by the Brier score is lost among the huge amount of correctly predicted 0s."), to justify further the specific focus on the BLI.
* * *
Minor comments:

line 41 reference WCS(2021) looks weird. What is WCS ?
* * *
Thank you for pointing this out. We now refer to the World Climate Service website by "(World-Climate-Service, 2021)".